# On Integrated Clustering and Outlier Detection

**Lionel Ott**
University of Sydney
lott4241@uni.sydney.edu.au

**Linsey Pang**
University of Sydney
qlinsey@it.usyd.edu.au

**Fabio Ramos**
University of Sydney
fabio.ramos@sydney.edu.au

**Sanjay Chawla**
University of Sydney
sanjay.chawla@sydney.edu.au

## Abstract

We model the joint clustering and outlier detection problem using an extension of the facility location formulation. The advantages of combining clustering and outlier selection include: (i) the resulting clusters tend to be compact and semantically coherent (ii) the clusters are more robust against data perturbations and (iii) the outliers are contextualised by the clusters and more interpretable. We provide a practical subgradient-based algorithm for the problem and also study the theoretical properties of algorithm in terms of approximation and convergence. Extensive evaluation on synthetic and real data sets attest to both the quality and scalability of our proposed method.

## 1 Introduction

Clustering and outlier detection are often studied as separate problems [1]. However, it is natural to consider them simultaneously. For example, outliers can have a disproportionate impact on the location and shape of clusters which in turn can help identify, contextualize and interpret the outliers. Pelillo [2] proposed a game theoretic definition of clustering algorithms which emphasis the need for methods that require as little information as possible while being capable of dealing with outliers.

The area of "robust statistics" studies the design of statistical methods which are less sensitive to the presence of outliers [3]. For example, the median and trimmed mean estimators are less sensitive to outliers than the mean. Similarly, versions of Principal Component Analysis (PCA) have been proposed [4, 5, 6] which are more robust against model mis-specification and outliers. An important primitive in the area of robust statistics is the notion of Minimum Covariance Determinant (MCD): Given a set of $n$ multivariate data points and a parameter $\ell$, the objective is to identify a subset of points which minimizes the determinant of the variance-covariance matrix over all subsets of size $n - \ell$. The resulting variance-covariance matrix can be integrated into the Mahalanobis distance and used as part of a chi-square test to identify multivariate outliers [7].

In the theoretical computer science literature, similar problems have been studied in the context of clustering and facility location. For example, Chen [8] has considered and proposed a constant factor approximation algorithm for the k-median with outliers problem: Given $n$ data points and parameters $k$ and $\ell$, the objective is to remove a set of $\ell$ points such that the cost of k-median clustering on the remaining $n - \ell$ points is minimized. Our model is similar to the one proposed by Charikar et. al. [9] who have used a primal-dual formulation to derive an approximation algorithm for the facility location with outlier problem.

More recently, Chawla and Gionis [10] have proposed $k$-means--, a practical and scalable algorithm for the k-means with outlier problem. k-means-- is a simple extension of the $k$-means algorithm and is guaranteed to converge to a local optima. However, the algorithm inherits the weaknesses of the

classical $k$-means algorithm. These are: (i) the requirement of setting the number of clusters $k$ and (ii) initial specification of the $k$ centroids. It is well known that the choice of $k$ and initial set of centroids can have a disproportionate impact on the result.

In this paper we model clustering and outlier detection as an integer programming optimization task and then propose a Lagrangian relaxation to design a scalable subgradient-based algorithm. The resulting algorithm discovers the number of clusters and requires as input: the distance (discrepancy) between pairs of points, the cost of creating a new cluster and the number $\ell$ of outliers to select.

The remainder of the paper is structured as follows. In Section 2 we formally describe the problem as an integer program. In Section 3, we describe the Lagrangian relaxation and details of the subgradient algorithm. The approximation properties of the relaxation and the convergence of the subgradient algorithm are discussed in Section 4. Experiments on synthetic and real data sets are the focus of Section 5 before concluding with Section 6. The supplementary section derives an extension of the affinity propagation algorithm [11] to detect outliers (APOC) - which will be used for comparison.

## 2  Problem Formulation

The Facility Location with Outliers (**FLO**) problem is defined as follows [9]. Given a set of data points with distances $D = \{d_{ij}\}$, the cluster creation costs $c_i$ and the number of outliers $\ell$, we define the task of clustering and outlier detection as the problem of finding the assignments to the binary exemplar indicators $y_j$, outlier indicators $o_i$ and point assignments $x_{ij}$ that minimizes the following objective function:

$$\textbf{FLO} \equiv \min \sum_j c_j y_j + \sum_i \sum_j d_{ij} x_{ij}, \tag{1}$$

$$\text{subject to} \quad x_{ij} \leq y_j \tag{2}$$

$$o_i + \sum_j x_{ij} = 1 \tag{3}$$

$$\sum_i o_i = \ell \tag{4}$$

$$x_{ij}, y_j, o_i \in \{0, 1\}. \tag{5}$$

In order to obtain a valid solution a set of constraints have been imposed:

- points can only be assigned to valid exemplars Eq. (2);
- every point must be assigned to exactly one other point or declared an outlier Eq. (3);
- exactly $\ell$ outliers have to be selected Eq. (4);
- only integer solutions are allowed Eq. (5).

These constraints describe the facility location problem with outlier detection. This formulation will allow the algorithm to select the number of clusters automatically and implicitly defines outliers as those points whose presence in the dataset has the biggest negative impact on the overall solution.

The problem is known to be NP-hard and while approximation algorithms have been proposed, when distances are assumed to be a metric, there is no known algorithm which is practical, scalable, and comes with solution guarantees [9]. For example, a linear relaxation of the problem and a solution using a linear programming solver is not scalable to large data sets as the number of variables is $O(n^2)$. In fact we will show that the Lagrangian relaxation of the problem is exactly equivalent to a linear relaxation and the corresponding subgradient algorithm scales to large data sets, has a small memory footprint, can be easily parallelized, and does not require access to a linear programming solver.

## 3  Lagrangian Relaxation of FLO

The Lagrangian relaxation is based on the following recipe and observations: (i) relax (or dualize) "tough" constraints of the original **FLO** problem by moving them to the objective; (ii) associate

a Lagrange multiplier ($\lambda$) with the relaxed constraints which intuitively captures the price of constraints not being satisfied; (iii) For any non-negative $\lambda$, $\mathbf{FLO}(\lambda)$ is a lower-bound on the $\mathbf{FLO}$ problem. As a function of $\lambda$, $\mathbf{FLO}(\lambda)$ is a concave but non-differentiable; (iv) Use a subgradient algorithm to maximize $\mathbf{FLO}(\lambda)$ as a function of $\lambda$ in order to close the gap between the primal and the dual.

More specifically, we relax the constraint $o_i + \sum_j x_{ij} = 1$ for each $i$ and associate a Lagrange multiplier $\lambda_i$ with each constraint. Rearranging the terms yields:

$$\mathbf{FLO}(\lambda) = \min \underbrace{\sum_i (1 - o_i)\lambda_i}_{\text{outliers}} + \underbrace{\sum_j c_j y_j + \sum_i \sum_j (d_{ij} - \lambda_i) x_{ij}}_{\text{clustering}}. \tag{6}$$

$$\text{subject to} \quad x_{ij} \leq y_i \tag{7}$$

$$\sum_i o_i = \ell \tag{8}$$

$$0 \leq x_{ij}, y_j, o_i \in \{0, 1\} \quad \forall i, j \tag{9}$$

We can now solve the relaxed problem with a heuristic finding valid assignments that attempt to minimize Eq. (6) without optimality guarantees [12]. The Lagrange multipliers $\lambda$ act as a penalty incurred for constraint violations which we try to minimize. From Eq. (6) we see that the penalty influences two parts: outlier selection and clustering. The heuristic starts by selecting good outliers by designating the $\ell$ points with largest $\lambda$ as outliers, as this removes a large part of the penalty. For the remaining $N - \ell$ points clustering assignments are found by setting $x_{ij} = 0$ for all pairs for which $d_{ij} - \lambda_i \geq 0$. To select the exemplars we compute:

$$\mu_j = c_j + \sum_{i: d_{ij} - \lambda_i < 0} (d_{ij} - \lambda_i), \tag{10}$$

which represents the amortized cost of selecting point $j$ as exemplar and assigning points to it. Thus, if $\mu_j < 0$ we select point $j$ as an exemplar and set $y_j = 1$, otherwise we set $y_j = 0$. Finally, we set $x_{ij} = y_j$ if $d_{ij} - \lambda_i < 0$. From this complete assignment found by the heuristic we compute a new subgradient $\mathbf{s}^t$ and update the Lagrangian multipliers $\lambda^t$ as follows:

$$\mathbf{s}_i^t = 1 - \sum_j x_{ij} - o_i \tag{11}$$

$$\lambda_i^t = \max(\lambda_i^{t-1} + \theta_t \mathbf{s}_i, 0), \tag{12}$$

where $\theta_t$ is the step size at time $t$ computed as

$$\theta_t = \theta_0 \, \text{pow}(\alpha, t) \quad \alpha \in (0, 1), \tag{13}$$

where $\text{pow}(a, b) = a^b$. To obtain the final solution we repeat the above steps until the changes become small enough, at which point we extract a feasible solution. This is guaranteed to converge if a step function is used for which the following holds [12]:

$$\lim_{n \to \infty} \sum_{t=1}^n \theta_t = \infty \quad \text{and} \quad \lim_{t \to \infty} \theta_t = 0. \tag{14}$$

A high level algorithm description is given in Algorithm 1.

## 4   Analysis of Lagrangian Relaxation

In this section, we analyze the solution obtained from using the Lagrangian relaxation (LR) method. Our analysis will have two parts. In the first part, we will show that the Lagrangian relaxation is exactly equivalent to solving the linear relaxation of the $\mathbf{FLO}$ problem. Thus if FLO(IP), FLO(LP) and FLO(LR) are the optimal value of integer program, linear relaxation and linear programming solution respectively, we will show that FLO(LR) = FLO(LP). In the second part, we will analyze the convergence rate of the subgradient method and the impact of outliers.

**Algorithm 1:** LagrangianRelaxation()

---

Initialize $\lambda^0, \mathbf{x}^0, t$
**while** *not converged* **do**
    $\mathbf{s}^t \leftarrow$ ComputeSubgradient($\mathbf{x}^{t-1}$)
    $\lambda^t \leftarrow$ ComputeLambda($\mathbf{s}^t$)
    $\mathbf{x}^t \leftarrow \mathbf{FLO}(\lambda^t)$      (solve via heuristic)
    $t \leftarrow t + 1$
**end**

---

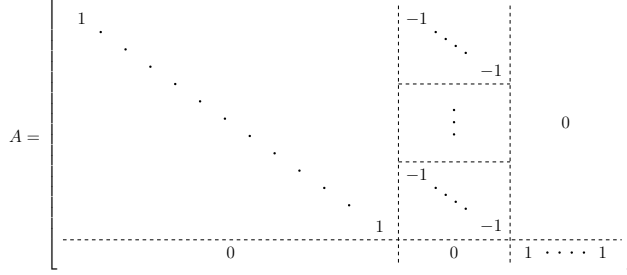

Figure 1: Visualization of the building blocks of the $A$ matrix. The top left is a $n^2 \times n^2$ identity matrix which is followed by $n$ row stacked blocks of $n \times n$ negative identity matrices. To the right of those is another $n^2 \times n$ block of zeros. The final row in the block matrix consists of $n^2 + n$ zeros followed by $n$ ones.

## 4.1 Quality of the Lagrangian Relaxation

Consider the constraint set $L = \{(x, y, o) \in \mathbb{Z}^{n^2+2n} | x_{ij} \leq y_j \wedge \sum_i o_i \leq \ell \; \forall \; i, j\}$. Then it is well known that the optimal value of FLO(LR) of the Lagrangian relaxation is equal to the cost of the following optimization problem [12]:

$$\min \sum_j c_j y_j + \sum_i \sum_j x_{ij} d_{ij} \tag{15}$$

$$o_i + \sum_j x_{ij} = 1 \tag{16}$$

$$conv(L) \tag{17}$$

where $conv(L)$ is the convex hull of the set $L$. We now show that $L$ is integral and therefore

$$conv(L) = \{(x, y, o) \in \mathbb{R}^{n^2+2n} | x_{ij} \leq y_j \wedge \sum_i o_i \leq \ell \; \forall \; i, j\}$$

This in turn will imply that FLO(LR) = FLO(LP). In order to show that $L$ is integral, we will establish that that the constraint matrix corresponding to the set $L$ is totally unimodular (TU). For completeness, we recall several important definitions and theorems from integer program theory [12]:

**Definition 1.** *A matrix $A$ is totally unimodular if every square submatrix of $A$, has determinant in the set $\{-1, 0, 1\}$.*

**Proposition 1.** *Given a linear program: $\min\{c^T x : Ax \geq b, x \in R_+^n\}$, let $b$ be the set of integer vectors for which the problem instance has finite value. Then the optimal solution has integral solutions if $A$ is totally unimodular.*

An equivalent definition of total unimodularity (TU) and often easier to establish is captured in the following theorem.

**Theorem 1.** *Let $A$ be a matrix. Then $A$ is TU iff for any subset of rows $X$ of $A$, there exists a coloring of rows of $X$, with 1 or -1 such that the weighted sum of every column (while restricting the sum to rows in $X$) is -1, 0 or 1.*

We are now ready to state and prove the main theorem in this section.

**Theorem 2.** *The matrix corresponding to the constraint set $L$ is totally unimodular.*

*Proof.* We need to consider the constraints

$$x_{ij} \leq y_j \ \forall \ i, j \tag{18}$$

$$\sum_{i=1}^{n} o_i \leq \ell \tag{19}$$

We can express the above constraints in the form $Au = b$ where $u$ is the vector:

$$u = [x_{11}, \ldots, x_{1n}, \ldots, x_{n1}, \ldots, x_{nn}, y_1, \ldots, y_n, o_1, \ldots, o_n]^T \tag{20}$$

The block matrix $A$ is of the form:

$$A = \begin{bmatrix} I & B & 0 \\ 0 & 0 & \mathbf{1} \end{bmatrix} \tag{21}$$

Here $I$ is an $n^2 \times n^2$ identity matrix, $B$ is stack of $n$ matrices of size $n \times n$ where each element of the stack is a negative identity matrix, and $\mathbf{1}$ is an $1 \times n$ block of $1's$. See Figure 1 for a detailed visualization.

Now to prove that A is TU, we will use Theorem 1. Take any subset $X$ of rows of $A$. Whether we color the rows of $X$ by 1 or -1, the column sum (within $X$) of a column of $I$ will be in $\{-1, 0, 1\}$. A similar argument holds for columns of the block matrix $\mathbf{1}$. Now consider the submatrix $B$. We can express $X$ as

$$X = \cup_{i=1, i \in B(X,:)}^{n} X_i \tag{22}$$

where each $X_i = \{r \in X | X(r, i) = -1\}$. Given that $B$ is a stack of negative diagonal matrices, $X_i \cap X_j = \emptyset$ for $i \neq j$. Now consider a column $j$ of $B$. If $X_j$ has even number of $-1's$, then split the elements of $X_j$ evenly and color one half as 1 and the other as $-1$. Then the sum of column $j$ (for rows in $X$) will be 0. On the other hand, if another set of rows $X_k$ has odd number of $-1$, color the rows of $X_k$ alternatively with 1 and $-1$. Since $X_j$ and $X_k$ are disjoint their colorings can be carried out independently. Then the sum of column $j$ will be 1 or $-1$. Thus we satisfy the condition of Theorem 1 and conclude that $A$ is TU. $\qquad \square$

## 4.2 Convergence of Subgradient Method

As noted above, the langrangian dual is given by $\max\{\mathbf{FLO}(\lambda) | \lambda \geq \mathbf{0}\}$. Furthermore, we use a gradient ascent method to update the $\lambda$'s as $[\lambda_i^t]_{i=1}^{n} = \max(\lambda_i^{t-1} + \theta_t \mathbf{s}_i, 0)$ where $\mathbf{s}_i^t = 1 - \sum_j x_{ij} - o_i$ and $\theta_t$ is the step-size.

Now, assuming that the norm of the subgradients are bounded, i.e., $\|\mathbf{s}\|_2 \leq G$ and the distance between the initial point and the optimal set, $\|\lambda_1 - \lambda^*\|_2 \leq R$, it is known that [13]:

$$|Z(\lambda^t) - Z(\lambda^*)| \leq \frac{R^2 + G^2 \sum_{i=1}^{t} \theta_i^2}{2 \sum_{i=1}^{t} \theta_i}$$

This can be used to show that to obtain $\epsilon$ accuracy (for any step size), the number of iterations is lower bounded by $O(RG/\epsilon^2)$, We examine the impact of integrating clustering and outliers on the convergence rate. We make the following observations:

**Observation 1.** *At a given iteration $t$ and for a given data point $i$, if $o_i^t = 1$ then $\sum_j x_{ij}^t = 0$ and $s_i^t = 0$ and therefore $\lambda_i^{t+1} = \lambda_i^t$.*

**Observation 2.** *At a given iteration $t$ and for a given data point $i$, if $o_i^t = 0$ and the point $i$ is assigned to exactly one exemplar, then $\sum_j x_{ij}^t = 1$ and therefore $s_i^t = 0$ and $\lambda_i^{t+1} = \lambda_i^t$.*

In conjunction with the algorithm for solving FLO($\lambda$) and the above observations we can draw important conclusions regarding the behavior of the algorithm including (i) the $\lambda$ values associated with outliers will be relatively larger and stabilize earlier and (ii) the $\lambda$ values of the exemplars will be relatively smaller and will take longer to stabilize.

# 5 Experiments

In this section we evaluate the proposed method on both synthetic and real data and compare it to other methods. We first present experiments using synthetic data to show quantitative analysis of the methods in a controlled environment. Then, we present clustering and outlier results obtained on the MNIST image data set.

We compare our Langrangian Relaxation (LR) based method to two other methods, k-means-- and an extension of affinity propagation [11] to outlier clustering (APOC) whose details can be found in the supplementary material. Both LR and APOC require a cost for creating clusters. We obtain this value as $\alpha * \text{median}(d_{ij})$, i.e. the median of all distances multiplied by a scaling factor $\alpha$ which typically is in the range $[1, 30]$. The initial centroids required by k-means-- are found using k-means++ [14] and unless specified otherwise k-means-- is provided with the correct number of clusters $k$.

## 5.1 Synthetic Data

We use synthetic datasets for controlled performance evaluation and comparison between the different methods. The data is generated by randomly sampling $k$ clusters with $m$ points, each from $d$-dimensional normal distributions $\mathcal{N}(\mu, \Sigma)$ with randomly selected $\mu$ and $\Sigma$. To these clusters we add $\ell$ additional outlier points that have a low probability of belonging to any of the selected clusters. The distance between points is computed using the Euclidean distance. We focus on 2D distributions as they are more challenging then higher dimensional data due to the separability of the data.

To assess the performance of the methods we use the following three metrics:

1. Normalized Jaccard index, measures how accurately a method selects the ground truth outliers. It is a coefficient computed between selected outliers $O$ and ground-truth outliers $O^*$. The final coefficient is normalized with regards to the best possible coefficient obtainable in the following way:

$$J(O, O^*) = \frac{|O \cap O^*|}{|O \cup O^*|} / \frac{\min(|O|, |O^*|)}{\max(|O|, |O^*|)}. \tag{23}$$

2. Local outlier factor [15] (LOF) measures the outlier quality of a point. We compute the ratio between the average LOF of $O$ and $O^*$, which indicates the quality of the set of selected outliers.

3. V-Measure [16] indicates the quality of the overall clustering solution. The outliers are considered as an additional class for this measure.

For the Jaccard index and V-Measure a value of $1$ is optimal, while for the LOF factor a larger value is better.

Since the number of outliers $\ell$, required by all methods, is typically not known exactly we explore how its misspecification affects the results. We generate 2D datasets with 2000 inliers and 200 outliers and vary the number of outliers $\ell$ selected by the methods. The results in Figure 2 show that in general none of the methods fail completely if the value of $\ell$ is misspecified. Looking at the Jaccard index, which indicates the percentage of true outliers selected, we see that if $\ell$ is smaller then the true number of outliers all methods pick only outliers. When $\ell$ is greater then the true number of outliers we can see a that LR and APOC improve with larger $\ell$ while k-means-- does only sometimes. This is due to the formulation of LR which selects the largest outliers, which APOC does to some extent as well. This means that if some outliers are initially missed they are more likely to be selected if $\ell$ is larger then the true number of outliers. Looking at the LOF ratio we can see that selecting more outliers then present in the data set reduces the score somewhat but not dramatically, which provides the method with robustness. Finally, V-Measure results show that the overall clustering results remain accurate, even if the number of outliers is misspecified.

We experimentally investigate the quality of the solution by comparing with the results obtained by solving the LP relaxation using CPLEX. This comparison indicates what quality can be typically expected from the different methods. Additionally, we can evaluate the speed of these approximations. We evaluate 100 datasets, consisting of 2D Gaussian clusters and outliers, with varying number of

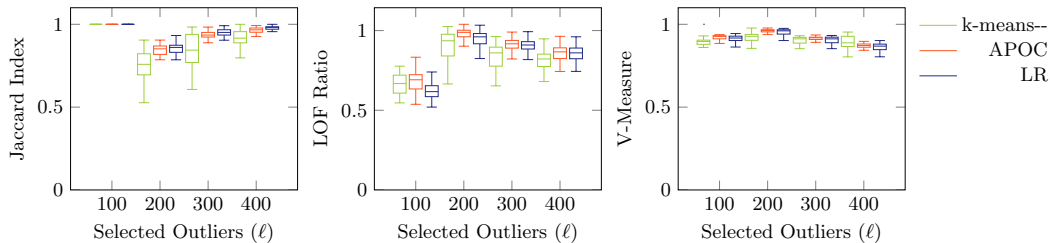

Figure 2: The impact of number of outliers specified ($\ell$) on the quality of the clustering and outlier detection performance. LR and APOC perform similarly with more stability and better outlier choices compared to k-means--. We can see that overestimating $\ell$ is more detrimental to the overall performance, as indicated by the LOF Ratio and V-Measure, then underestimating it.

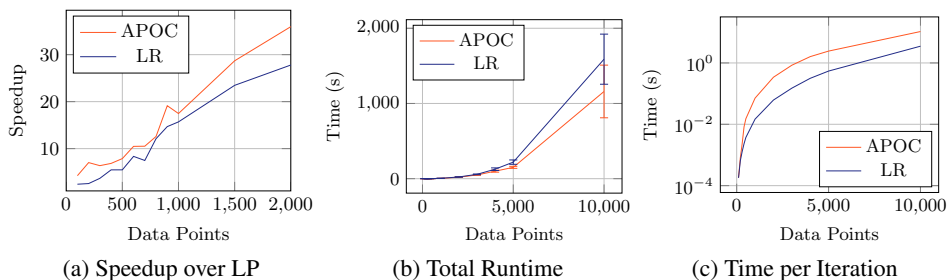

|(a) Speedup over LP|(b) Total Runtime|(c) Time per Iteration|

Figure 3: The graphs shows how the number of points influences different measures. In (a) we compare the speedup of both LR and APOC over LP. (b) compares the total runtime needed to solve the clustering problem for LR and APOC . Finally, (c) plots the time required (on a log scale) for a single iteration for LR and APOC.

points. On average LR obtains $94\% \pm 5\%$ of the LP objective value, APOC obtains an energy that is $95\% \pm 4\%$ of the optimal solution found by LP and k-means--, with correct k, obtains $86\% \pm 12\%$ of the optimum. These results reinforce the previous analysis; LR and APOC perform similarly while outperforming k-means--. Next we look at the speed-up of LR and APOC over LP. Figure 3 a) shows both methods are significantly faster with the speed-up increasing as the number of points increases. Overall for a small price in quality the two methods obtain a significantly faster solution. k-means-- outperforms the other two methods easily with regards to speed but has neither the accuracy nor the ability to infer the number of clusters directly from the data.

Next we compare the runtime of LR and APOC. Figure 3 b) shows the overall runtime of both methods for varying number of data points. Here we observe that APOC is faster then LR, however, by observing the time a single iteration takes, shown in Figure 3 c), we see that LR is much faster on a per iteration basis compared to APOC. In practice LR requires several times the number of iterations of APOC, which is affected by the step size function used. Using a more sophisticated method of computing the step size will provide large gains to LR. Finally, the biggest difference between LR and APOC is that the latter requires all messages and distances to be held in memory. This obviously scales poorly for large datasets. Conversely, LR computes the distances at runtime and only needs to store indicator vectors and a sparse assignment matrix, thus using much less memory. This makes LR amenable to processing large scale datasets. For example, with single precision floating point numbers, dense matrices and $10\,000$ points APOC requires around $2200\,\mathrm{MB}$ of memory while LR only needs $370\,\mathrm{MB}$. Further gains can be obtained by using sparse matrices which is straight forward in the case of LR but complicated for APOC.

## 5.2 MNIST Data

The MNIST dataset, introduced by LeCun et al. [17], contains $28 \times 28$ pixel images of handwritten digits. We extract features from these images by representing them as 768 dimensional vectors which is reduced to 25 dimensions using PCA. The distance between these vectors is computed using the $L2$ norm. In Figure 4 we show exemplary results obtained when processing $10\,000$ digits with the

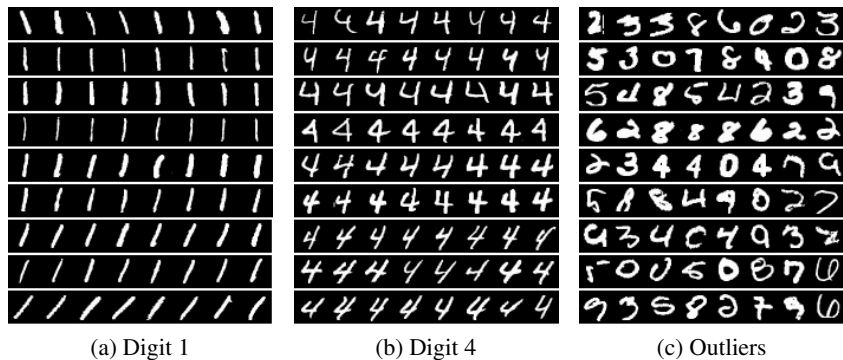

|              (a) Digit 1              |              (b) Digit 4              |              (c) Outliers              |

Figure 4: Each row in (a) and (b) shows a different appearance of a digit captured by a cluster. The outliers shown in (c) tend to have heavier then usual stroke, are incomplete or are not recognizable as a digit.

Table 1: Evaluation of clustering results of the MNIST data set with different cost scaling values $\alpha$ for LR and APOC as well as different settings for k-means--. We can see that increasing the cost results in fewer clusters but as a trade off reduces the homogeneity of the clusters.

|              |      | LR   |      | APOC | k-means-- |      |
|--------------|------|------|------|------|------|------|
| $\alpha$     | 5    | 15   | 25   | 15   | n.a. | n.a. |
| V-Measure    | 0.52 | 0.67 | 0.54 | 0.53 | 0.51 | 0.58 |
| Homogeneity  | 0.78 | 0.74 | 0.65 | 0.72 | 0.50 | 0.75 |
| Completeness | 0.39 | 0.61 | 0.46 | 0.42 | 0.52 | 0.47 |
| Clusters     | 120  | 13   | 27   | 51   | 10   | 40   |

LR method with $\alpha = 5$ and $\ell = 500$. Each row in Figure 4 a) and b) shows examples of clusters representing the digits 1 and 4, respectively. This illustrates how different the same digit can appear and the separation induced by the clusters. Figure 4 c) contains a subset of the outliers selected by the method. These outliers have different characteristics that make them sensible outliers, such as: thick stroke, incomplete, unrecognizable or ambiguous meaning.

To investigate the influence the cluster creation cost has we run the experiment with different values of $\alpha$. In Table 1 we show results for LR with values of cost scaling factor $\alpha = \{5, 15, 25\}$, APOC with $\alpha = 15$ and k-means-- with $k = \{10, 40\}$. We can see that LR obtains the best V-Measure score out of all methods with $\alpha = 15$. The homogeneity and completeness scores reflect this as well, while homogeneity is similar to other settings the completeness value is much better. Looking at APOC we see that it struggles to obtain the same quality as LR. In the case of k-means-- we can observed how providing the algorithm with the actual number of clusters results in worse performance compared to a larger number of clusters which highlights the advantage of methods capable of automatically selecting the number of clusters from the data.

## 6 Conclusion

In this paper we presented a novel approach to joint clustering and outlier detection formulated as an integer program. The method only requires pairwise distances and the number of outliers as input and detects the number of clusters directly from the data. Using a Lagrangian relaxation of the problem formulation, which is solved using a subgradient method, we obtain a method that is provably equivalent to a linear programming relaxation. Our proposed algorithm is simple to implement, highly scalable, and has a small memory footprint. The clusters and outliers found by the algorithm are meaningful and easily interpretable.

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
