[Supplementary Material]

# Supplementary Material
# On Integrated Clustering and Outlier Detection

## 1   Affinity Propagation Outlier Clustering

The extension to affinity propagation [1], based on the binary variable model [2], solves the integer program of Section 2 by representing it as a factor graph, shown in Figure 1. This factor graph is solved using belief propagation and is based on the following energy function:

$$\max \sum_{ij} S_{ij}(x_{ij}) + \sum_{j} E_j(x_{:j}) + \sum_{i} I_i(x_{i:}, o_{i:}) + \sum_{k} P_k(o_{:k}), \tag{1}$$

where

$$S_{ij}(x_{ij}) = \begin{cases} -c_i & \text{if } i = j \\ -d_{ij} & \text{otherwise} \end{cases} \tag{2}$$

$$I_i(x_{i:}, o_{i:}) = \begin{cases} 0 & \text{if } \sum_j x_{ij} + \sum_k o_{ik} = 1 \\ -\infty & \text{otherwise} \end{cases} \tag{3}$$

$$E_j(x_{:j}) = \begin{cases} 0 & \text{if } x_{jj} = \max_i c_{ij} \\ -\infty & \text{otherwise} \end{cases} \tag{4}$$

$$P_k(o_{:k}) = \begin{cases} 0 & \text{if } \sum_i o_{ik} = 1 \\ -\infty & \text{otherwise} \end{cases} \tag{5}$$

with $x_{i:} = x_{i1}, \ldots, x_{iN}$. Since we use the max-sum algorithm we maximise the energy function and use negative distances. The three constraints can be interpreted as follows:

1. 1-of-$N$ Constraint ($I_i$). Each data point has to choose exactly one exemplar or be declared as an outlier.

2. Exemplar Consistency Constraint ($E_j$). For point $i$ to select point $j$ as its exemplar, point $j$ must declare itself an exemplar.

3. Select $\ell$ Outliers Constraint ($P_k$). For every outlier selection exactly one point is assigned.

These constraints are enforced by associating an infinite cost with invalid configurations, thus resulting in an obviously suboptimal solution.

The energy function is optimised with the max-sum algorithm [3], which allows the recovery of the maximum a posteriori (MAP) assignments of the $x_{ij}$ and $o_{ik}$ variables. The algorithm works by exchanging messages between nodes in the factor graph. In their most general form these messages are defined as follows:

$$\mu_{v \rightarrow f}(x_v) = \sum_{f^* \in ne(v) \backslash f} \mu_{f^* \rightarrow v}(x_v), \tag{6}$$

$$\mu_{f \rightarrow v}(x_v) = \max_{x_1, \ldots, x_M} \left[ f(x_v, x_1, \ldots, x_M) + \sum_{v^* \in ne(f) \backslash v} \mu_{v^* \rightarrow f}(x_{v^*}) \right], \tag{7}$$

Figure 1: (a) Messages exchanged by the APOC graphical model. $x_{ij}$ represents the clustering choice whereas $o_{ik}$ represents the outlier choice. (b) Graphical model of APOC. The left part is responsible for the clustering of the data, while the right part is responsible for the outlier selection. These two parts interact with each other via the $I$ factor nodes.

---

**Algorithm 1:** apoc$(S, \ell)$

---

**foreach** $i, j \in \{1, \ldots, N\}$ **do**
  $\alpha_{ij} \leftarrow 0$;
  $\rho_{ij} \leftarrow 0$;
**end**
**foreach** $i \in \{1, \ldots, N\}, k \in \{1, \ldots, \ell\}$ **do**
  $\lambda_{ik} \leftarrow 0$;
  $\omega_{ik} \leftarrow \text{median}(S)$;
**end**
**repeat**
  update $\rho$ according to Eq. (8);
  update $\alpha$ according to Eq. (9);
  update $\lambda$ according to Eq. (10);
  update $\omega$ according to Eq. (11);
**until** *convergence*;
$O \leftarrow$ extract outliers;
$E \leftarrow$ extract exemplars;
$A \leftarrow$ find exemplar assignments;

---

where $\mu_{v \to f}(x)$ is the message sent from node $v$ to factor $f$, $\mu_{f \to v}(x_v)$ is the message from factor $f$ sent to node $v$, $ne()$ is the set of neighbours of the given factor or node and $x_v$ is the value of node $v$.

The messages exchanged by APOC are shown in Figure 1a. We can see that each node $x_{ij}$ is connected to three factors: $S_{ij}$, $I_i$ and $E_j$ whereas outlier nodes $o_{ik}$ are connected to only two, $I_i$ and $P_k$. Messages $\rho_{ij}, \beta_{ij}, \tau_{ik}$ and $\xi_{ik}$ are sent from nodes to factors and derived using Eq. (6). The other five messages $s_{ij}, \alpha_{ij}, \eta_{ij}, \lambda_{ik}$ and $\omega_{ik}$ are derived with Eq. (7) since they are sent from a factor to a node. Since only binary variables are involved it is sufficient to compute the difference between the two variable settings. Combining these messages we obtain the final set of update equations as:

$$\rho_{ij} = s_{ij} + \min\left[-\max_{t \neq j}(\alpha_{it} + s_{it}), -\max_t(\omega_{it})\right] \tag{8}$$

$$\alpha_{ij} = \begin{cases} \sum_{t \neq j} \max(0, \rho_{tj}) & i = j \\ \min\left[0, \rho_{jj} + \sum_{t \notin \{i,j\}} \max(0, \rho_{tj})\right] & i \neq j \end{cases} \tag{9}$$

$$\lambda_{ik} = \min\left[-\max_t(\alpha_{it} + s_{it}), -\max_{t \neq k}(\omega_{ti})\right] \tag{10}$$

$$\omega_{ik} = -\max_{t \neq i}(\lambda_{tk}) \tag{11}$$

The above equations show how to update the messages, however, we still need to explain how to initialise the messages, determine convergence and extract the MAP solution. First, it is important to set the diagonal entries of $S$ properly. Typically using $S_{ii} = \theta * median(S)$ is a good choice, with $\theta \in [1, 30]$. The messages $\alpha_{ij}, \rho_{ij}$ and $\lambda_{ik}$ are initialised to $0$ and $\omega_{ik}$ to the median of $S$. Once the messages are initialised we update them in turn with damping until we achieve convergence. Convergence is achieved when the energy of the solution is not changing drastically over a few iterations. The outliers are determined as the $\ell$ points with the largest values of $\max_k(\lambda_{ik} + \omega_{ik})$. From the remaining points the exemplars are then selected as the points for which $(\alpha_{ii} + \rho_{ii}) > 0$ is true. All other points $i$ are assigned to the exemplar $e$ satisfying $\arg\max_e(\alpha_{ie} + \rho_{ie})$. This entire process is shown in Algorithm 1, where we first initialise the messages, then update them until convergence and finally extract the MAP solution. The pseudo code in Algorithm 1 provides a conceptual overview of the steps performed during clustering.