[Reviews · NeurIPS 2014]

Submitted by Assigned_Reviewer_6

The paper looks at the problem of combining clustering and outlier detection. It is very well written and easy to read. The authors reuse an earlier facilities location without outliers formulation by Charikar et' al and their main contribution is the solution to the problem formulation.

The FLO problem was shown to be intractable by the authors of that paper and no approximation algorithm exists that is both i) scalable and ii) comes with guarantees. The FLO paper has well over 100 citations and a quick scan of them shows this to be case. The Lagrange relaxation the author's propose is straight forward and seems reasonable. The analysis of the LR is interesting, the core result that the LP relaxation of FLO is equivalent to the LR relaxation is non-intuitive.

The experimental results are a nice addition to the paper in that they show the usefulness of the formulation particularly the speed up over the LP relaxation. However, I found the lack of absolute values in the experimental section raised questions. For example in Figure 3, your results show the speedup over LP relaxation but doesn't show how fast the LR method is in absolute time.

Summary: The authors take an existing formulation (FLO) show how it is useful for outlier detection and clustering simultaneously. Their main contribution is the LR formulation for the existing objective function and the experimental results showing its usefulness.

Submitted by Assigned_Reviewer_13

This paper considers an important problem in unsupervised machine learning and optimization, clustering with outlier detection. Clustering has a long history in both theoretical and practical areas. People worked on many types of clustering problems before, such as k-means, k-medians and k-centers in geometric background, and correlation clustering, spectral clustering in graph theory. However, one critical issue for clustering is outlier detection, which could influence the final result significantly.

The authors present a gradient descent algorithm for clustering with outlier detection. They slightly modify the integer programing model from [8], through adding the outlier part to the constraint. Then they relax the model into a sequence of Lagrange relaxations, and solve it via a gradient descent strategy.

The experiment considers both of synthetic and real data, and shows the advantages over other two methods.

Positive points:

1. Using Lagrange relaxation is a new idea for outlier detection.

2. The algorithm is clean, and easy to implement, which makes it practical.

Negative points:

1. The theoretical analysis is not enough. For example, in section 4.2, the authors should provide more details for the convergence.

2. More references are needed. In computational statistics, there are many new techniques on trimming outlier for regression and clustering, such as David Mount et al, ``A practical approximation algorithm for the LMS line estimator" and ``On the least trimmed squares estimator".
Summary: This paper provides a Lagrange relaxation approach for a hard problem in clustering area. The technique is new, but needs more theoretical analysis.

Submitted by Assigned_Reviewer_22

The paper describes an approach to clustering which deals automatically with the problem of outlier detection. The whole idea is based on the so-called Facility Location with Outliers (FLO) problem, introduced and studied in theoretical computer science in 2001 (see ref. [8] in the paper). The novelty of the proposed approach lies mainly in the study ot the Lagrangian relaxation of the original (integer) linear programming problem. In particular, the authors show that the Lagrangian relaxation is equivalent to solving the linear relaxation of FLO, and also analyze the convergence properties of the subgradient method used, which makes the algorithm highly scalable. Some experiments on both synthetic and real-world data sets (MINST) are presented.

The paper is well written and organized, it’s easy to follow and understand, despite many technicalities, and its motivations and goals are quite clear.

Also, I like the idea of introducing, within the machine learning community, this apparently novel formulation of the clustering problem which, to my knowledge, was originally confined only to the theoretical computer science domain. In regards to the statement made by the authors that “clustering and outlier detection are often studied as separate problems,” however, I’d like to draw the authors’ attention to novel clustering formulations which, abandoning the idea of partitioning the input data set, focus instead on the notion of a cluster itself (see, e.g., M. Pelillo, “What is a cluster?” NIPS’09 Workshop on Clustering, and references therein). In fact, in these approaches, which are non-partitional by definition, one can see that the problem of clustering and outlier detetion are simply two faces of the same coin.

One problem with the proposed formulation, though, is that while it doesn’t need as input the number of clusters, which is of course good, it neverthless needs to know the number of outliers, which is even more problematic than knowing the number of clusters. Also, it needs to know the “cost” of creating a new cluster, and here, again, it’s not clear how to define them in practical applications (the choice made in the experiments reported here look quite heuristic).

My final comment concerns the experimental validation. Indeed, the results presented here do not show a clear and substantial improvement over the other methods used. This is particularly clear in the MNIST experimemts (Table 1) which show that APOC and LR are basically equivalent, and k-means-- is better than the proposed LR algorithm.

Minor comments:

- Definition 2 is indeed a Proposition
Summary: Although the experimental results are not compelling and the method requires a problematic setting of parameters, I think the ideas proposed in this paper are potentially interesting and to some extent novel. Hence, I would be inclined to give the authors the opportunity to present their work at NIPS in the poster format (urging them to possibly provide more experimental evidence of the effectiveness of the approach).
Author Feedback
Author rebuttal: We thank the reviewers for their thoughtful comments and address their comments below.

Assigned_Reviewer_13

Q1: The theoretical analysis is not enough. For example, in section 4.2, the authors should provide more details for the convergence.

A1: The standard subgradient convergence result is that to obtain \epsilon accuracy, the number of iterations required is O(1/\epsilon^{2}). The result, as we note, applies for the FLO problem. Furthermore, we have observed (Observation 1 and 2 in Section 4.2), that if data has genuine outliers, then the \lambda values for outliers stabilize much earlier (in the iteration process). However, it is easy to create data sets, for example ones with no genuine outliers, where these observations will not hold. Thus the conclusion is that convergence is at least as good as maximizing a concave piecewise linear function.

Q2. More references are needed. In computational statistics, there are many new techniques on trimming outlier for regression and clustering, such as David Mount et al, ``A practical approximation algorithm for the LMS line estimator" and ``On the least trimmed squares estimator".

A2: The above references fall under the area known as “robust statistics” which we alluded to in [2]. We have looked at them and they are certainly relevant and we will definitely add them in the next revision.

Assigned_Reviewer_22

Q1: I’d like to draw the authors’ attention to novel clustering formulations which, abandoning the idea of partitioning the input data set, focus instead on the notion of a cluster itself (see, e.g., M. Pelillo, “What is a cluster?” NIPS’09 Workshop on Clustering, and references therein

A1: We read Pelillo’s paper and it was very interesting to note that our approach satisfies four of the six features that he proposes a clustering solution should fulfil: (i) no assumption on the underlying data representation: we only require an affinity matrix; (ii) no assumption on the structure of the affinity matrix: our problem formulation places no restriction on the affinity matrix; (iii) no restriction on number of clusters: we also place no restriction but we don’t extract them sequentially: (iv) It leaves clutter elements unassigned: this is what our proposed method does but we call clutter elements outliers.

Q2: One problem with the proposed formulation, though, is that while it doesn’t need as input the number of clusters, which is of course good, it nevertheless needs to know the number of outliers, which is even more problematic than knowing the number of clusters. Also, it needs to know the “cost” of creating a new cluster, and here, again, it’s not clear how to define them in practical applications (the choice made in the experiments reported here look quite heuristic).

A2: Since the real number of outliers is typically unknown we investigate the impact that the misspecification of the number of outliers has on the results, which is shown in Figure 2. From our experiments LR and APOC are not severely affected by selecting more than the true number of outliers.

We think think that it should be possible to remove the need to specify the need for the number of outliers by formulating the initial problem (Equation 1) differently, i.e. assigning a cost when declaring a point an outlier as opposed to the currently fixed number of outliers. However, this would add another parameter similar to the cluster creation cost one..

The “cost” associated with cluster selection uses a simple heuristic:
factor * median(distance_matrix)
where the factor is typically somewhere between 5 to 30. While this is a tunable parameter it is defined in terms of the distance matrix.

Q3: My final comment concerns the experimental validation. Indeed, the results presented here do not show a clear and substantial improvement over the other methods used. This is particularly clear in the MNIST experiments (Table 1) which show that APOC and LR are basically equivalent, and k-means-- is better than the proposed LR algorithm.

A3: We have since investigated the results of the MNIST dataset and by using a better step-size function the LR method outperforms the other two versions which had no change in performance, even when attempting to tweak their parameters further. We state the best results with the other two methods for reference again below.

LR LR LR APOC k-means--
alpha 5 15 25 15 n.a.
V-Measure 0.52 0.67 0.54 0.53 0.58
Homogeneity 0.78 0.74 0.65 0.72 0.75
Completeness 0.39 0.61 0.46 0.42 0.47
Clusters 120 13 27 51 40

Q4: Definition 2 is indeed a Proposition

A4: Fixed, thank you.

Assigned_Reviewer_6

Q1: The experimental results are a nice addition to the paper in that they show the usefulness of the formulation particularly the speed up over the LP relaxation. However, I found the lack of absolute values in the experimental section raised questions. For example in Figure 3, your results show the speedup over LP relaxation but doesn't show how fast the LR method is in absolute time.

A1: We’d like to clarify that the absolute runtime for both LR and APOC is indeed shown in Figure 3b.